# Hypoglycemic Effects and Quality Marker Screening of *Dendrobium nobile* Lindl. at Different Growth Years

**DOI:** 10.3390/molecules29030699

**Published:** 2024-02-02

**Authors:** Yi Luo, Da Yang, Yanzhe Xu, Di Wu, Daopeng Tan, Lin Qin, Xingdong Wu, Yanliu Lu, Yuqi He

**Affiliations:** 1Key Lab of the Basic Pharmacology of The Ministry of Education, Zunyi Medical University, 6 West Xue-Fu Road, Zunyi 563009, China; luoyi0902@foxmail.com (Y.L.); da_yang@yeah.net (D.Y.); yanzhe.pharma@foxmail.com (Y.X.); 2Guizhou Engineering Research Center of Industrial Key-Technology for Dendrobium Nobile, Zunyi Medical University, 6 West Xue-Fu Road, Zunyi 563009, China; wd_32677@126.com (D.W.); tandp@zmu.edu.cn (D.T.); qinlin1115@163.com (L.Q.); wuxingdong@zmu.edu.cn (X.W.)

**Keywords:** *Dendrobium nobile* Lindl., growth year, diabetic, quality markers

## Abstract

(1) Background: The effect of *Dendrobium nobile* Lindl. (*D. nobile*) on hyperglycemic syndrome has only been recently known for several years. Materials of *D. nobile* were always collected from the plants cultivated in various growth ages. However, regarding the efficacy of *D. nobile* on hyperglycemic syndrome, it was still unknown as to which cultivation age would be selected. On the other hand, with the lack of quality markers, it is difficult to control the quality of *D. nobile* to treat hyperglycemic syndrome. (2) Methods: The effects of *D. nobile* cultivated at year 1 and year 3 were checked on alloxan-induced diabetic mice while their body weight, diet, water intake, and urinary output were monitored. Moreover, levels of glycosylated serum protein and insulin were measured using Elisa kits. The constituents of *D. nobile* were identified and analyzed by using UPLC-Q/trap. Quality markers were screened out by integrating the data from UPLC-Q/trap into a network pharmacology model. (3) Results: The *D. nobile* cultivated at both year 1 and year 3 showed a significant effect on hyperglycemic syndrome at the high dosage level; however, regarding the significant level, *D. nobile* from year 1 showed the better effect. In *D. nobile*, most of the metabolites were identified as alkaloids and sesquiterpene glycosides. Alkaloids, represented by dendrobine, were enriched in *D. nobile* from year 1, while sesquiterpene glycosides were enriched in *D. nobile* from year 3. Twenty one metabolites were differentially expressed between *D. nobile* from year 1 and year 3. The aforementioned 21 metabolites were enriched to 34 therapeutic targets directly related to diabetes. (4) Conclusions: Regarding the therapy for hyperglycemic syndrome, *D. nobile* cultivated at year 1 was more recommended than that at year 3. Alkaloids were recommended to be used as markers to control the quality of *D. nobile* for hyperglycemic syndrome treatment.

## 1. Introduction

Diabetes mellitus (DM) is a metabolic disorder characterized by disturbances in the metabolism of sugars, fats, proteins, water, and electrolytes due to an absolute or relative lack of insulin secretion and/or its utilization disorders, with hyperglycemia as its primary symptom. According to the World Health Organization, China has the highest number of DM patients in the world, and its prevalence is increasing annually [1]. Serials of chemical drugs to treat hyperglycemia are available, such as iminosugars, sugar derivatives, and metformin [2,3,4]. Among them, metformin is the most recommended drug for type 2 diabetes (T2D) treatment [5]. However, the effect of metformin is still not 100% guaranteed and still leads to obvious side effects. At least 30% of patients, receiving metformin, reported adverse effects, including diarrhea, nausea, and vomiting, with their underlying mechanisms poorly understood [6]. Thus, exploring a new therapeutic strategy is always on the way. Traditional Chinese medicines (TCMs) were used for thousands of years in eastern Asia such as China, Japan, and Korea. There is a representative syndrome called “Xiao Ke Disease”, which shares the same symptoms as DM [7]. Thus, the traditional treatment strategy for DM in TCM deserves attention. Studies showed that herbs enriching alkaloids [8,9] and polysaccharides [10,11] may possess better effects on controlling blood glucose levels.

*Dendrobium nobile* Lindl. (*D. nobile*), commonly known as Jin Chai Shi Hu, is a perennial epiphytic herb of the *Orchidaceae* family, mainly distributed in Guizhou, Sichuan, Yunnan of China [12]. *D. nobile* is the main source of medicinal dendrobium and has a long history of use in traditional Chinese medicine, which could be traced back to the Shan Hai Jing period. In the Chinese Pharmacopoeia, it has the ability to nourish the stomach and Yin, produce saliva and antipyresis, and is used in febrile diseases accompanied by fluid lack, thirst and worry, deficiency of stomach Yin, decreased food intake accompanied by vomiting, and continuous hyperpyrexia after chronic illness. In modern research, *D. nobile* is effective in reducing blood glucose levels, regulating glucose metabolism-related genes such as *Glut2*, *Glut4*, *FoxO1*, and *Pgc1a* [13], regulating gluconeogenesis processes, and improving insulin resistance [14]. It may also have hypoglycemic effects by protecting against oxidative damage in the liver and kidneys [15]. 

The hypoglycemic effect of *D. nobile* has also been preliminarily verified in clinical observations [16]; its clinical needs are developing rapidly. However, the establishment of quality standards for *D. nobile* is incomplete, especially the pharmacological effects-oriented quality markers that are needed. This situation limits the clinical use of *D. nobile* due to its efficacy not being clearly defined. In the 2020 edition of the *Chinese Pharmacopoeia*, only the content of dendrobine (Den, C_16_H_25_NO_2_) not being less than 0.40% was used as the quality evaluation standard [17], which is obviously not comprehensive enough. Building up new overall assessment standards for the quality of *D. nobile* will ease this challenge.

As reported in a large amount of studies in the literature, the chemical composition of *D. nobile* is rather complex, including alkaloids [18], sesquiterpenes [19], flavonoids [20], phenolic acids, polysaccharides [21], etc. But the main chemical substance for its hypoglycemic effect has yet to be determined [15]. Previous studies from our group have used ultra-high performance liquid chromatography-quadrupole/time of flight mass spectrometry (UPLC-Q/TOF-MS) to perform comparative analyses for secondary metabolites of *D. nobile* at different growth years and in different growing environments. The results showed that the content of the alkaloids and sesquiterpene glycosides in *D. nobile* is related to its growth year. The one-year-old *D. nobile* has a relatively higher level of alkaloids, while the three-year-old *D. nobile* has a higher level of sesquiterpene glycosides. Therefore, different growth years will have a significant impact on the overall profile of secondary metabolites in *D. nobile*, and this further suggests that there may be pharmacological activity differences among *D. nobile* at different growth ages. 

Therefore, this study will first determine the differences in secondary metabolites in *D. nobile* at different growth years. Secondly, a diabetic mouse model induced by alloxan [22,23] will be used to evaluate the hypoglycemic effect of *D. nobile* at different growth years. Finally, the overall profile of secondary metabolites of *D. nobile* will be analyzed using UPLC-QE/Orbitrap/MS. Differential components will be screened and identified, and their effects on key targets related to diabetes will be predicted using network pharmacology in order to preliminarily sort out the potential quality markers.

## 2. Results

### 2.1. Changes in the Contents of Dendrobine and Six Sesquiterpene Glycosides in D. nobile at Different Growth Years

Figure 1 shows the structures of dendrobine and six sesquiterpene glycosides in *D. nobile*. The contents of dendrobine and six bibenzyls were determined, and the results revealed that the content of dendrobine was significantly higher in the one-year-old *D. nobile* than in the three-year-old one (*p* < 0.05). Conversely, the levels of sesquiterpene glycosides, including Dn A, Dn C, Dn D, D G, Dm D, and Dn E, were significantly higher in three-year-old *D. nobile* than the other (*p* < 0.05), as shown in Figure 2. These findings demonstrate that there are differences in the chemical compositions of *D. nobile* at different growth stages, providing a basis for screening substances with superior pharmacological effects in subsequent studies.

### 2.2. Comparison of the Hypoglycemic Effects of D. nobile at Different Growth Years

#### 2.2.1. Animal Modeling and Grouping Results of Diabetes Induced by Alloxan

After the induction with alloxan, a total of 108 mice had a blood glucose level ≥ 11.1 mmol/L, indicating that the modeling was successful. These mice were randomly divided into the model control and treatment groups, while the mice receiving saline were randomly selected in the normal control group (n = 12). Compared to the control group, significant statistical differences in blood glucose and body weight were observed in all the groups (*p* < 0.01), as shown in Figure 3.

#### 2.2.2. The Impact of *D. nobile* of Different Growth Years on Blood Glucose and Body Weight in Diabetic Mice

During the administration period, the effects of different growth-year *D. nobile* on fasting blood glucose and body weight of diabetic mice were observed weekly, and the results are shown in Figure 4 and Figure 5. It can be learned that compared to the model group, in the first week of administration, the high-dose one-year-old *D. nobile* already showed a significant hypoglycemic effect (*p* < 0.05 or 0.01), and in the fourth week of administration, the blood glucose level of the medium-dose one-year-old group mice significantly decreased (*p* < 0.05). In addition, compared to the model group, after four weeks of administration, the high-dose one-year-old *D. nobile* showed a significant recovery effect on body weight (*p* < 0.01).

#### 2.2.3. The Impact of Different Growth Years of *D. nobile* on the Overall Situation of Diabetic Mice

During the administration period, the overall situation of the mice was observed daily. It was found that the mice of the control group were in good condition, with glossy fur and agile reactions. The diabetic model group mice were listless, tended to curled up together, and had reduced activity with messy fur. The glossiness of the fur and the mental state of the mice in each treatment group showed improvement. Information about the diet, water intake, and urine output of mice was collected and recorded. The results are shown in Figure 6. Compared with the model group, the high-dose one-year-old *D. nobile* group showed a significant improvement (*p* < 0.05) on diet, water intake, and urine volume. Combined with the effect on body weight, high-dose one-year-old *D. nobile* can significantly improve typical symptoms of diabetes, such as polydipsia, polyuria, polyphagia, and weight loss.

#### 2.2.4. The Effects of *D. nobile* with Different Growth Years on the Pancreatic Tissue of Diabetic Mice Observed by H&E Staining

The H&E staining results shown in Figure 7 reveal an intact pancreatic structure with clear boundaries and an orderly arrangement of islet cells in control group mice. In contrast, the islet cells of the model group were smaller, accompanied by an irregular morphology, increased intercellular space, a severe reduction in cell number, and disordered tissue arrangement, indicating cell degeneration. Administering the drug visibly improved islet morphology and restored the cell numbers of the islets, indicating a certain protective effect against the damage caused by alloxan on pancreatic tissue. 

#### 2.2.5. The Effect of Different Growth Periods of *D. nobile* on Glucose Tolerance in Diabetic Mice

As shown in Figure 8, compared to the model group, there was a statistical difference in the area under the curve of the first-year medium- and high-dose groups (*p* < 0.05) after 4 weeks of administration. The glucose tolerance levels of the mice were significantly restored, which means that the administration of *D. nobile* effectively improves the ability of regulating blood glucose.

#### 2.2.6. The Effect of Different Growth Periods of *D. nobile* on Biochemical Indicators of Diabetic Mice

Due to the fact that serum protein synthesis is faster than hemoglobin synthesis (with a half-life of approximately 20 days), glycosylated serum protein was selected as the observation indicator that reflected the blood glucose changes during the past 1–3 weeks. As shown in Figure 9A, compared to the model group, all the groups that received *D. nobile* treatment, except for the low-dose group of the three-year-old *D. nobile*, significantly reduced the glycosylated serum protein levels of diabetic mice (*p* < 0.05). Compared to the high-dose group of the three-year-old *D. nobile*, the high dose of the one-year-old *D. nobile* had better blood glucose-lowering effects (*p* < 0.05). As shown in Figure 9B, the restoration of fasting insulin level by *D. nobile* did not reach statistical significance.

#### 2.2.7. The effect of Different Growth Periods of *D. nobile* on Insulin Resistance and Pancreatic Islet β Function

HOMA-IR at a high level indicates that the body needs more insulin to maintain blood glucose balance, and the higher value of HOMA-IR means a worse situation of insulin resistance; HOMA-β at a low level indicates poor compensation function of the pancreatic tissue, and as the value increases, the pancreatic function is enhanced. The outcomes of the calculation for the two indices are shown in Figure 10. High-dose, one-year-old *D. nobile* effectively improved insulin resistance in mice (*p* < 0.05) and restored their pancreatic islet β function (*p* < 0.01).

### 2.3. Preliminary Screening of Quality Markers of D. nobile

#### 2.3.1. Fingerprint of Secondary Metabolites of *D. nobile*

As shown in Figure 11, significant differences were observed in certain peaks when comparing the one-year-old and three-year-old *D. nobile* samples, which means that with the extension of growth years, the secondary metabolite profile of *D. nobile* changed.

#### 2.3.2. Similarity Evaluation

The similarity of the fingerprint of each batch of *D. nobile* was measured by comparing it with a reference fingerprint map generated by the Chinese Medicine Chromatographic Fingerprinting Similarity Evaluation System (2012 edition) software, and the outcomes revealed that the similarity was between 0.840 and 0.957, indicating that the overall profile was similar. However, there were significant differences in the response of some peaks, as shown in Figure 12 and Figure 13.

#### 2.3.3. Principal Component Analysis of The Main Components of The Secondary Metabolites of *D. nobile* with Different Growth Ages

Principal component analysis (PCA) was conducted on the peak areas of the fingerprint spectra of 12 batches of *D. nobile* samples. The results showed a clear separation between the one-year-old and three-year-old *D. nobile* samples, as shown in Figure 14. Each point in the figure represents a sample, with the x, y, and z axes representing the scores of each sample projected onto the principal components PC1, PC2, and PC3, respectively. The model’s goodness is indicated by the fitting parameters within a 95% confidence interval, with R2X = 0.901 and Q2 = 0.841. 

#### 2.3.4. OPLS-DA Analysis of Secondary Metabolites of *D. nobile* with Different Growth Years

To further investigate the differences in secondary metabolites of *D. nobile* with different growth years, OPLS-DA discriminant analysis was conducted. The results showed that the model had R2X = 0.972 and Q2 = 0.531 (R2X is closer to 1, which indicates that the model is more stable, and Q2 > 0.4 indicates an acceptable model) without overfitting. The OPLS-DA results were presented in a 2D scatter plot, as shown in Figure 15A. One-year-old and three-year-old *D. nobile* were clearly separated into two regions. Based on the criterion of variable importance in projection (VIP) > 1, a total of 229 secondary metabolites were selected, as shown in Figure 15B. 

#### 2.3.5. Identification of Different Components of *D. nobile* with Different Growth Ages

For the secondary metabolites with VIP > 1, a total of 21 compounds were identified based on their accurate molecular weight, secondary mass spectra fragmentation information, *m*/*z* Cloud database, and literature reports [24,25,26]. The results are shown in Table 1. The secondary mass spectra and fragmentation patterns can be found in Appendix A.

#### 2.3.6. Disease Enrichment of Different Growth-Year *D. nobile* with Differential Components

Twenty-one differential components with known structures were subjected to target prediction using the Swiss Target Prediction database. The probability value ranges from zero to one, which assesses the likelihood of the target prediction results of the database, and the value closer to one indicates a higher accuracy. A filtering condition of probability values greater than zero was used to screen the duplicates among the 192 targets. Using the WebGestalt database (https://www.webgestalt.org/, accessed on: 21 January 2023), disease enrichment analysis was performed on the 192 targets. The prediction of 40 diseases was ranked in ascending order of *p*-value, and the top 15 important diseases were visualized in a disease enrichment bubble plot, as shown in Figure 16. These diseases mainly included metabolic diseases and neurological diseases, among which diseases directly associated with diabetes were Diabetes Mellitus, Experimental (C0011853), Diabetes Mellitus, Non-Insulin-Dependent (C0011860), and Insulin Resistance (C0021655).

#### 2.3.7. Differential Component–Target Interactions of *D. nobile* with Different Growth Years

Thirty-four target genes related to diabetes were identified. The predicted targets of 21 secondary metabolites contained all 34 target genes, indicating that the hypoglycemic effect of *D. nobile* is associated with these 21 differential components. The corresponding component–target interaction network was constructed and visualized using Cytoscape software (Version 3.10.0), as shown in Figure 17, which includes 55 nodes and 125 edges. The purple circles represent differential components, while the pink hexagons represent the corresponding target genes. Each edge represents the interaction relationship between the chemical component and the target gene.

## 3. Discussion

It was reported that *Dendrobium* species showed an anti-hyperglycemia effect [27]. However, nobody discussed the plant source, especially the cultivation ages. Without restricting the cultivation ages, the conclusion may come with a certain level of bias. According to the data presented in [28], regarding anti-hyperglycemia effects, four Dendrobium species have a decreased order: *Dendrobium huoshanense*, *Dendrobium nobile*, *Dendrobium officinale*, and *Dendrobium chrysotoxum*. However, in the present study, we demonstrated that the effect of *D. nobile* depends on the cultivation ages. Thus, it is still needed to be further investigated as to which *Dendrobium* species should be used as recommended as a potent anti-hyperglycemia agent.

We have previously reported that *Dendrobium* alkaloids regulate the FXR-mediated signaling pathways as well as the PPARα-mediated signaling pathways [29,30]. In the present study, we proved that the alkaloids in *D. nobile* showed a higher correlation with *D. nobile* effect than other kinds of constituents, which means that FXR or PPARα may be key factors participating in the mechanisms by which *D. nobile* regulates blood glucose.

An ancient Chinese medicine book of importance is named “ShenLong BenCao Jing”. In this book, *D. nobile* was classified as one of the top-level herbs. In this classification, top-level herbs were proved to be effective and safe. However, in most cases, *D. nobile* was cultivated at year 3 or more than year 3. In particular, for *D. nobile* cultivated at year 1, there was no toxicology investigation, which is worth studying in the future.

It is well known that the quality control of Chinese medicine is essential for promoting the high-quality development of the traditional Chinese medicine industry and realizing the modernization of Chinese medicine [31]. Establishing a systematic, comprehensive, and scientific evaluation system to ensure clinical safety and effective administration is not only a research hotspot but also a research difficulty [32]. Considering that the chemical composition of Chinese medicine is hardly complex and diverse, it is also influenced by various factors, such as origin, age, harvesting season, planting method, and processing technology [33,34,35]. The viewpoint that single characteristic components cannot comprehensively evaluate the quality of Chinese medicine is widely accepted. The 2010 version of the *Chinese Pharmacopoeia* included characteristic fingerprint mapping for evaluating the quality of Chinese medicinal materials for the first time. This evaluation criteria system undoubtedly played a promising role in promoting and improving the quality management of Chinese medicine and the development of building quality standards for Chinese medicine. However, although the fingerprint mapping system largely characterizes the chemical information of Chinese medicine, it also has its obvious shortcomings. Finding pharmacologically oriented components or multi-index components to establish a quality standard system that conforms to the characteristics of Chinese traditional medicine is extremely urgent. Therefore, this study aims to find potent samples of *D. nobile* based on their hypoglycemic efficacy, screen its quality markers, and provide a foundation for establishing a scientific and rational evaluation system. 

Several clinical studies show that different extracts/sites of *D. nobile* have been used to reduce fasting plasma glucose (FBG), human hemoglobin A1c (HbA1c), and glycosylated serum protein (GSP), regulate dyslipidemia, improve antioxidant capacity, repair islet injuries, improve insulin resistance, etc. *D. nobile* has also been used to reduce the negative effects of oxidative stress on the liver and kidney tissues of diabetic mice by increasing the activities of the antioxidant enzymes SOD, CAT, and GSH. In addition to direct hypoglycemic effects, *D. nobile* also improves diabetic complications such as diabetic nephropathy and diabetic retinopathy [36].

The secondary metabolites of *D. nobile* are the material basis for its medicinal effect. The preliminary results of our group indicate that the accumulation of alkaloids and sesquiterpene glycosides, the secondary metabolites of *D. nobile*, varies significantly depending on the growth period, with the most significant difference observed between one-year-old and three-year-old *D. nobile* [7]. And the other report revealed that a significant shift in the composition of the fungal community within *D. nobile* stems was observed along the age gradient, mirroring dendrobine content changes [37]. This study further demonstrates that the content of dendrobine, an alkaloid, is significantly higher in one-year-old *D. nobile* than in the three-year-old specimens (*p* < 0.05), while the content of sesquiterpene glycosides, such as Dn A, Dn C, Dn D, DG, Dm D, and Dn E, is significantly higher in three-year-old *D. nobile* than in the one-year-old specimens (*p* < 0.05). Therefore, this study selected one-year-old and three-year-old *D. nobile* as research subjects and searched for a superior sample with a better hypoglycemic effect in a diabetic mouse model induced by alloxan. The results show that a high-dose extract of one-year-old *D. nobile* can significantly reduce fasting blood glucose, alleviate the symptoms of the three highs and one low, improve glucose metabolism, improve islet morphology, increase the number of cells in the islet interior, lower the level of serum glycated serum protein, improve insulin resistance, and enhance the function of islet β cells, which is, overall, superior to three-year-old *D. nobile.* Research has shown that changes in the contents of secondary metabolites are likely to have a significant impact on their biological activities [38,39]. This strongly suggests that the changes in secondary metabolites caused by different growth periods of *D. nobile* can be compared and analyzed to explore the material basis of its medicinal effect and screen for quality markers.

This study collected 12 batches of different growth periods of one-year- and three-year-old *D. nobile*. A reference fingerprint was established using the Chinese Medicine Chromatographic Fingerprint Similarity Evaluation System (2012 edition), and the similarity of each sample ranged from 0.840 to 0.957, indicating a similar overall profile but significant differences in the response of certain peaks. The PCA results showed that one-year-old and three-year-old *D. nobile* were clearly divided into two groups, and 229 small molecular compounds with VIP > 1 were screened based on OPLS-DA. Through analyzing the information, including accurate molecular weight, secondary fragment ion information, databases, and literature reports, 21 secondary metabolites were identified, including eleven lipids, four alkaloids, two terpenes, one phenolic acid, one sugar alcohol, one lignin, and one other compound. Predictive analysis for the target points of the 21 secondary metabolites not only revealed the enrichment of diseases directly related to diabetes but also all actions on the 34 target points directly related to diabetes. The results indicate that the 21 differential components are associated with the hypoglycemic effects of *D. nobile*. Moreover, these differential components have been reported in the chemical studies of *Dendrobium* species [40,41,42]. Among them, dendrobine is a characteristic component of *D. nobile* [43], and it has been reported to effectively relieve gestational diabetes in mice by reducing the secretion of inflammatory cytokines by T helper 17 (Th17) cells [44]. Our study found that the content of dendrobine in one-year-old *D. nobile* was higher than that in three-year-old *D. nobile*, which may be one of the reasons why one-year-old *D. nobile* is a superior sample for hypoglycemic purposes. 

In the differential component–target interaction network, the Degree value represents the number of connected pathways to a node, with higher Degree values indicating greater importance in the network. In this study, the four targets with the highest Degree values are PTPN1 (Protein Tyrosine Phosphatase Non-Receptor Type 1), HSD11B1 (Hydroxysteroid 11-Beta Dehydrogenase 1), CYP19A1 (Cytochrome P450 Family 19 Subfamily A Member 1), and PPARα. PTPN1 is a negative regulator of the insulin-PI3K/Akt signaling pathway [45] and is considered a potential therapeutic target for metabolic syndrome, obesity, and diabetes [46]. It is reported that an upregulation of PTPN1 can reverse glucose uptake capacity in gestational diabetic patients, improving their insulin resistance [47]. PTPN1 can also regulate food intake and energy expenditure by inhibiting leptin signaling in the hypothalamus, and a hypothalamus-specific loss of PTPN1 exacerbates diet-induced obesity in female mice [48]. HSD11B1 is highly expressed in abdominal adipose tissue, and its polymorphisms are closely related to obesity, metabolic syndrome, and diabetes [49,50]. Multiple studies demonstrate that HSD11B inhibitors decrease blood glucose levels in diabetic patients, increasing insulin sensitivity and improving insulin secretion disorders [51,52,53]. CYP19A1 is a potential target for diabetes treatment and can play an important role in this process. Research shows that CYP19A1 can convert androstenedione into estrogen, testosterone, and estradiol [54]. An estrogen deficiency is closely related to insulin resistance and subsequent damage in the pancreas, liver, muscles, and adipose tissue. Sulindac can treat diabetes in mice by activating CYP19A1 to synthesize estrogen [55], and multiple clinical data show that the CYP19A1 gene deeply affects plasma zinc levels, urinary zinc levels, susceptibility, and other issues in Chinese type 2 diabetic populations [56,57]. PPARα is mainly expressed in metabolically active tissues and is not only used for treating dyslipidemia [58] but also for regulating glucose metabolism [59], maintaining vascular endothelial function balance in diabetes [60], protecting diabetic patients’ visual nerves, inhibiting oxidative stress, anti-inflammatory, anti-diabetic kidney fibrosis, promoting neural repair, improving neural microcirculation, and reducing vascular regeneration injury, all of which play an important role in the chronic complications of diabetes [61,62,63]. The action network diagram in this study suggests that the mechanism of *D. nobile* in reducing blood glucose levels may be related to these targets, which also conforms to the characteristics of traditional Chinese medicine with multi-component and multi-target effects.

## 4. Materials and Methods

### 4.1. Plant Materials and Pre-Treatment

The samples of *D. nobile* were collected from various planting areas in the Chishui region of Guizhou province in China. They were identified as fresh stems of *D. nobile* by Professor Faming Wu from the school of pharmacy at Zunyi Medical University, according to the Chinese Pharmacopeia. Detailed information on the collection of *D. nobile* samples can be found in Table 2. The freshly harvested stems of *D. nobile* were separated, washed, air-dried, weighed for fresh weight, cut into sections of 1–2 cm, dried at 60 °C, and pulverized after being sieved through a 50-mesh sieve.

After pre-processing, the *D. nobile* samples were accurately weighed and added to a 95% ethanol–water solution. The mixture was then heated and refluxed three times, filtered while hot, concentrated at low pressure, and subsequently subjected to vacuum freeze-drying before storing.

### 4.2. Determination of Dendrobine Content by LC-MS

For this experiment, we precisely weighed the freeze-dried powder, diluted it with methanol, and centrifuged it to obtain the supernatant for testing, with pseudoephedrine as the internal standard. The determination of dendrobine content was performed on the SCIEX LC-MS system, using a Poroshell 120 EC-C18 column (3.0 × 100 mm, 2.7 μm) as the stationary phase and a gradient elution with 0.1% formic acid water (A)–acetonitrile (B) as the mobile phase. The elution conditions are described in detail in Table 3. The flow rate was set at 0.3 mL/min, the column temperature was maintained at 30 °C, and the injection volume was 2 μL. The mass spectrometry employed the positive ion multiple reaction monitoring (MRM) mode with electrospray ionization mass spectrometry (ESI) ionization. The ion spray voltage (IS) was set at 5500 V, the curtain gas (CUR) was set at 25 psi, the ion source gas 1 (GS1) was set at 40 psi, the ion source gas 2 (GS2) was set at 45 psi, and the temperature (TEM) was set at 450 °C. The collision gas (CAD) was set at 8 psi. The collision energy for dendrobine was 55 V, and for pseudophedrine, it was 15 V. The results of the methodological review are shown in the Appendix A.

### 4.3. Determination of the Content of Sesquiterpene Glycosides by LC-MS

For this procedure, we precisely weighed the freeze-dried powder, diluted it with methanol, and centrifuged to obtain the supernatant for testing. Longdanxieganin was used as the internal standard (final concentration of 100 ng/mL). We accurately weighed 5 mg of DnA, DnC, DnD, DG, DmD, and DnE reference substances, respectively, diluted them with methanol, and prepared a mixed reference solution of each reference substance at a concentration of 0.5 μg/mL. 

The content determination was performed on a SCIEX LC-MS using a Poroshell 120 EC-C18 (3.0 × 100 mm, 2.7 μm) chromatographic column as the stationary phase and water–acetonitrile as the mobile phase with a gradient elution at a flow rate of 0.3 mL/min and a column temperature of 30 °C. The injection volume was set to 2 μL. The mass spectrometry was conducted in negative ion MRM mode with ESI ionization using nitrogen as a nebulizer gas, with the ionization voltage set at −5.5 kV and the ion transfer tube temperature set at 450 °C.

### 4.4. Comparison of the Hypoglycemic Effects of D. nobile at Different Years of Growth

#### 4.4.1. Modeling and Administration

A total of 132 male C57BL/6J mice, aged 7–8 weeks and weighing 20–22 g, were purchased from Slac Jingda Experimental Animal Co., Ltd. (Changsha, Hunan, China). This experiment was approved by the Animal Experiment Ethics Committee of Zunyi Medical University (license number: [2021] 2-606). The mice were bred under the SPF condition, with the temperature maintained at 21–25 °C and humidity kept at 50–60%. The lighting condition followed a 12-h light/dark cycle, and the mice had ad libitum access to water and food during the breeding period. After a one-week adaptation period, all mice were randomly divided into a normal control group and an alloxan model group. After a 12-h fasting period without water deprivation, the model group received a tail intravenous injection of a 60 mg/kg solution of alloxan for two consecutive days, while the normal group received an equivalent volume of physiological saline. After 72 h, blood samples were taken from the tail to measure fasting blood glucose (FBG). The criterion for successful modeling is the FBG value ≥ 11.1 mmol/L [64].

Except for the normal control group (CG), the mice that were successfully modeled were randomly divided into 8 groups, namely the diabetes model group (DMG), metformin group (MG, 200 mg/kg/day), low dose of the one-year *D. nobile* group (ODN-L, 1.8 g/kg/day), medium dose of the one-year *D. nobile* group (ODN-M, 3.6 g/kg/day), high dose of the one-year *D. nobile* group (ODN-H, 7.2 g/kg/day), low-dose of the three-year *D. nobile* group (TDN-L, 1.8 g/kg/day), medium dose of the three-year *D. nobile* group (TDN-M, 3.6 g/kg/day), and high-dose of the three-year *D. nobile* group (TDN-H, 7.2 g/kg/day). The low dosage was equal to the indication in the Chinese Pharmacopeia [17]. The DMG and CG mice were orally administered the same volume (10 mL/kg) of 0.4% CMC-Na solution. Treatment continued for 4 weeks.

During the period of administration, the physical and mental states of mice were observed and recorded daily, while the fasting blood glucose (FBG) and body weight of the mice were measured and recorded weekly. Three days before the end of the administration, the food intake, water intake, and urine volume of the mice were collected and recorded through metabolic cages.

#### 4.4.2. Intraperitoneal Glucose Tolerance Test

One day prior to administration, the animals were fasted for 14 h. The intraperitoneal glucose tolerance test (IPGTT) was performed by injecting glucose at a dose of 2 g/kg [65]. Blood glucose levels were measured at 0, 30, 60, 90, and 180 min, and the area under the curve (AUC) was calculated.

#### 4.4.3. Biological Sample Collection

After drug administration, anesthesia was performed, and eyeballs were removed to collect blood. The animals were euthanized by cervical dislocation. Pancreatic tissue was obtained and washed with physiological saline; approximately 2–3 cm of tissue from the tail of the pancreas was collected and stored in a 10% neutral formaldehyde solution. The remaining tissue was rapidly frozen in liquid nitrogen and then transferred to a −80 °C freezer for storage.

#### 4.4.4. Pancreatic Tissue Hematoxylin and Eosin Staining

Pancreatic tissue was fixed in a 10% neutral methanal solution for 48 h. After progressive dehydration in different concentrations of ethanol, it was soaked in xylene for 3 h and embedded in 5-µm-thick paraffin slices after being immersed in paraffin. The paraffin slices were baked at 60 °C for 40 min and then soaked in xylene twice for 10 min each time. Subsequently, the slices were subjected to dewaxing in anhydrous ethanol and 80% ethanol in turn, and then stained with hematoxylin for 15 min. After being washed with water for 30 s, the slices were stained with eosin for 2 min and then washed with water for 10 min. The slices were dehydrated twice and soaked in xylene for 5 min before being taken out and sealed with neutral resin for observation under an upright microscope.

#### 4.4.5. Measurement of Biochemical Indicators

Whole blood was placed in a 1.5-mL centrifuge tube and kept at 4 °C for 30 min. It was then centrifuged at 4 °C and 4500 rpm for 15 min to separate the serum. The glycosylated serum protein (GSP) and fasting insulin (FINS) levels in the serum were measured according to the instructions.

Based on the FINS and FBG levels, the Homeostatic Model Assessment of Insulin Resistance (HOMA-IR) was calculated using a steady-state model to evaluate the insulin resistance index. The calculation formula is as follows:HOMA-IR = FBG × FINS/22.5 (1)

According to the FINS and FBG levels, the Homeostatic Model Assessment of Beta-cell function (HOMA-β) can be evaluated using a steady-state model, and the calculation formula is as follows:HOMA-β = 20 × FINS/(FBG-3.5) (2)

### 4.5. Screening of Secondary Metabolites with Differential Expression in D. nobile

#### 4.5.1. Sample Preparation

We precisely weighed 1 g of *D. nobile* samples from different batches (detailed information is shown in Table 2), and added a 10-fold volume of 95% ethanol–aqueous solution. The mixture was heated under reflux at 85 °C for 2 h each time, repeated 3 times. The filtrates were combined and concentrated under reduced pressure to the same volume to ensure the same content of crude drugs. 

#### 4.5.2. Chromatography Parameters

The UPLC-QE/Orbitrap/MS was used to analyze the secondary metabolites of *D. nobile*. The analysis was performed on a Hypersil GOLD C18 column (2.1 × 150 mm, 1.9 μm) as the stationary phase, with a mobile phase consisting of 0.1% formic acid solution (A) and acetonitrile (B). The gradient elution conditions are shown in Table 4, and the detection time ranged from 3.5 to 68 min. The flow rate was set at 0.3 mL/min, and the column temperature was maintained at 40 °C. A sample volume of 3 μL was injected for this analysis.

#### 4.5.3. Mass Spectrometry Parameters

The mass spectrometry technique used electrospray ionization (ESI) in positive ion mode. It employed a two-step scanning mode, with a full scan at first and automatic triggering of secondary mass spectrometry scans. The scan range was set from 50 to 1500 Da. The electrospray voltage was maintained at 3.5 kV, and the ionization temperature was set at 300 °C. The sheath gas flow rate was set at 35 arb, and the auxiliary gas flow rate was set at 15 arb. The capillary temperature was maintained at 350 °C, and the collision energy was set at 25 V. The results of the methodological review are shown in the Appendix A.

#### 4.5.4. Similarity Evaluation

Twelve batches of *D. nobile* powder samples (S1 to S12) were taken in an appropriate amount. Using the Pharmaceutical Chromatographic Fingerprint Similarity Evaluation System (2012 edition), with S1 as the reference sample and a time window width set at 0.5 min, overlapping fingerprint profiles were generated through multipoint calibration and automatic peak matching. The median method was used to generate the reference fingerprint profile (R) for *D. nobile*, and the similarity of the fingerprint profiles of the 12 batches of *D. nobile* and the reference profile were evaluated.

#### 4.5.5. Multivariate Statistical Analysis

Compound Discover (3.2) software was used to perform the peak matching, peak alignment, ion fusion, and deconvolution processing of the raw data from mass spectrometry detection. The MS tolerance and MS2 tolerance were set to 20 and 20 ppm, respectively. The common peak area data from 12 batches of *D. nobile* samples were grouped using Simca-p (14.1.0) software. Unsupervised pattern recognition principal component analysis (PCA) and supervised orthogonal partial least squares discriminant analysis (OPLS-DA) were performed sequentially, with a variable importance projection (VIP) greater than 1 used as a screening condition, and scatter plots were generated.

#### 4.5.6. Identification of Differential Components

By combining the precise molecular weights of the differential components with the first and second levels of mass spectrometry, false-positive fragmentation peaks were excluded. Compound Discover (3.2) software was used for fitting calculations, followed by matching with the reference literature and the *m*/*z* Cloud database to preliminarily infer the structure. 

### 4.6. Prediction of Quality Biomarkers by Network Pharmacology

The PubChem and ChemSpider databases were used to search for the 3D structures of differential ingredients. The Swiss Target Prediction database was used to predict the target points. The predicted targets were screened with the condition of probability > 0, and the duplicated values were removed. WebGestalt was used for the disease enrichment analysis of the predicted targets, with the species limited to *Homo sapiens*. The Benjamini–Hochberg method was used for multiple comparisons with a set FDR of <0.05. The disease enrichment analysis results were visualized using the ggplot2 package in R 4.1.1 software to draw a bubble chart. Cytoscape 3.10.0 software was used to link the predominant components with the target points and draw the corresponding component–target interaction network diagram.

### 4.7. Statistical Methods

In this study, all the data were analyzed using SPSS software (version 29.0), and the results were presented as the means ± SEM. The independent sample *t*-test in this software was used to compare the differences between two groups, while the one-way ANOVA was used to compare differences between groups. When the variance was homogenous, the LSD test was employed, whereas the Dunnett test was used when the variance was not homogeneous. *p*-values of less than 0.05 were considered significant.

## 5. Conclusions

Regarding the therapy for hyperglycemic syndrome, *D. nobile* cultivated at year 1 were more recommended than that at year 3. Alkaloids were recommended to be used as the markers to control the quality of *D. nobile* for hyperglycemic syndrome treatment.

## Figures and Tables

**Figure 1 molecules-29-00699-f001:**
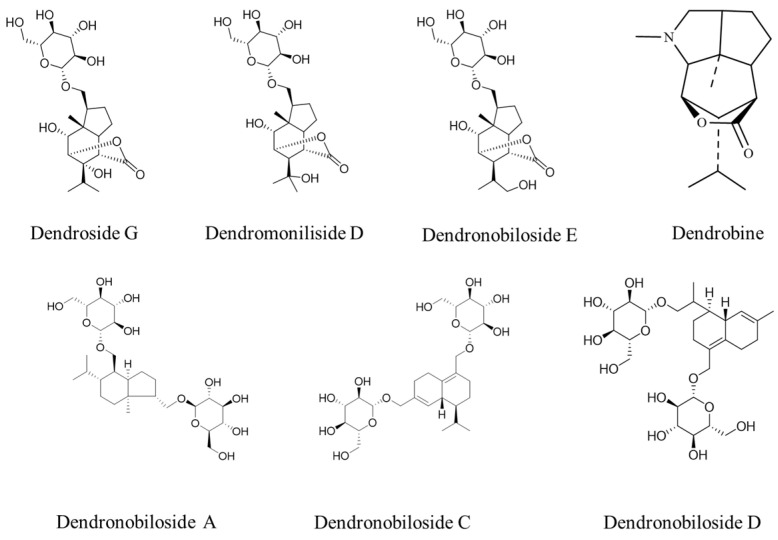
The structures of dendrobine and sesquiterpene glycosides.

**Figure 2 molecules-29-00699-f002:**
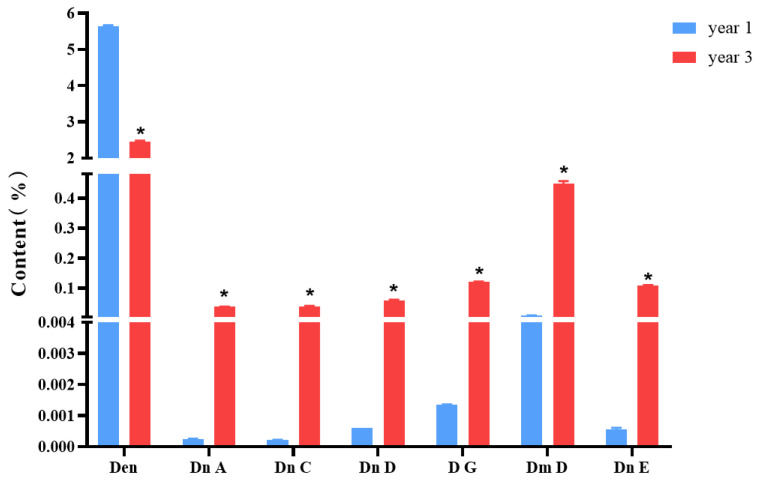
Changes in the contents of dendrobine and six sesquiterpene glycosides in *D. nobile* at different growth years (n = 3). The blue columns in the picture represent 1-year-old *D. nobile*, and the red columns represent 3-year-old *D. nobile*. Den: dendrobine; Dn A: dendronobiloside A; Dn C: dendronobiloside C; Dn D: dendronobiloside D; D G: dendroside G; Dm D: dendromoniliside D; and Dn E: dendronobiloside E. * represented *p* < 0.05 compared to the content of this compound in the extract of 1-year-old *D. nobile*.

**Figure 3 molecules-29-00699-f003:**
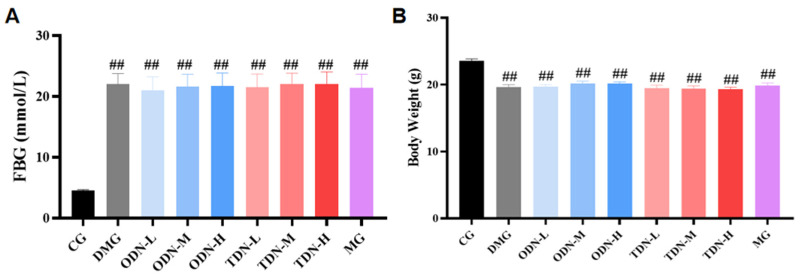
The blood glucose levels and body weights of the mice in each group after modeling (n = 12). (**A**) The bar graph of the fasting blood glucose level of each group after modeling; (**B**) the bar graph of body weights of each group after modeling. CG: control group; DMG: diabetes model group; ODN-L: low dose of the one-year-old *D. nobile* group (1.8 g/kg/day); ODN-M: middle dose of the one-year-old *D. nobile* group (3.6 g/kg/day); ODN-H: high dose of the one-year-old *D. nobile* group (7.2 g/kg/day); TDN-L: low dose of the three-year-old *D. nobile* group (1.8 g/kg/day); TDN-M: middle dose of the three-year-old *D. nobile* group (3.6 g/kg/day); and TDN-H: high dose of the three-year-old *D. nobile* group (7.2 g/kg/day). ## represents *p <* 0.01 compared to the control group.

**Figure 4 molecules-29-00699-f004:**
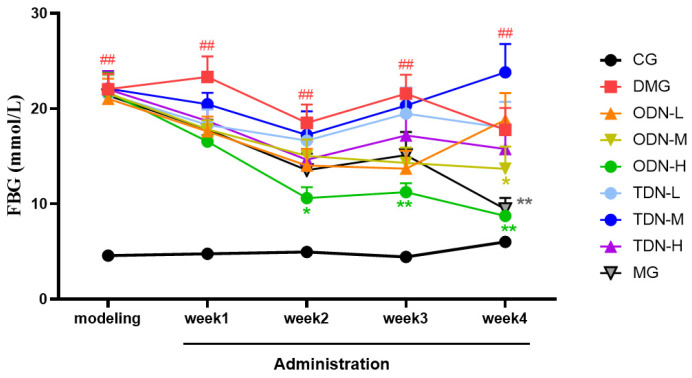
Blood glucose changes in each group of mice during drug administration (n = 12). The black curve represents the control group, the red curve represents the model group, the orange curve represents ODN-L, the yellow curve represents ODN-M, the green curve represents ODN-H, the cyan curve represents TDN-L, the blue curve represents TDN-M, the purple curve represents TDN-H, and the gray curve represents the positive drug group. CG: control group; DMG: diabetes model group; ODN-L: low dose of the one-year-old *D. nobile* group (1.8 g/kg/day); ODN-M: middle dose of the one-year-old *D. nobile* group (3.6 g/kg/day); ODN-H: high dose of the one-year-old *D. nobile* group (7.2 g/kg/day); TDN-L: low dose of the three-year-old *D. nobile* group (1.8 g/kg/day); TDN-M: middle dose of the three-year-old *D. nobile* group (3.6 g/kg/day); and TDN-H: high dose of the three-year-old *D. nobile* group (7.2 g/kg/day). ## represents *p <* 0.01 compared to the control group, * represents *p <* 0.05 compared to the model group, and ** represents *p <* 0.01 compared to the model group.

**Figure 5 molecules-29-00699-f005:**
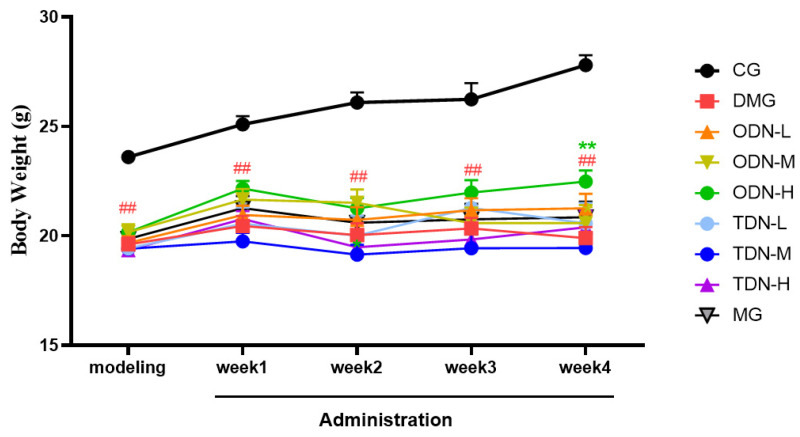
Body weight changes in each group of mice during drug administration (n = 12). The black curve represents the control group, the red curve represents the model group, the orange curve represents ODN-L, the yellow curve represents ODN-M, the green curve represents ODN-H, the cyan curve represents TDN-L, the blue curve represents TDN-M, the purple curve represents TDN-H, and the gray curve represents the positive drug group. CG: control group; DMG: diabetes model group; ODN-L: low dose of the one-year-old *D. nobile* group (1.8 g/kg/day); ODN-M: middle dose of the one-year-old *D. nobile* group (3.6 g/kg/day); ODN-H: high dose of the one-year-old *D. nobile* group (7.2 g/kg/day); TDN-L: low dose of the three-year-old *D. nobile* group (1.8 g/kg/day); TDN-M: middle dose of the three-year-old *D. nobile* group (3.6 g/kg/day); and TDN-H: high dose of the three-year-old *D. nobile* group (7.2 g/kg/day). ## represents *p <* 0.01 compared to the control group; ** represents *p <* 0.01 compared to the model group.

**Figure 6 molecules-29-00699-f006:**
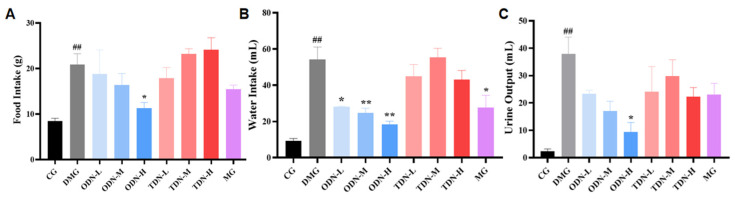
The effect of different growth years of *D. nobile* on the diet, water intake, and urine output of diabetic mice (n = 12). (**A**) Bar graph for the amount of food intake by each group of mice in the last three days of administration; (**B**) bar graph for the amount of water intake by each group of mice in the last three days of administration; (**C**) bar graph for the amount of urine produced by each group of mice in the last three days of administration. CG: control group; DMG: diabetes model group; ODN-L: low dose of the one-year-old *D. nobile* group (1.8 g/kg/day); ODN-M: middle dose of the one-year-old *D. nobile* group (3.6 g/kg/day); ODN-H: high dose of the one-year-old *D. nobile* group (7.2 g/kg/day); TDN-L: low dose of the three-year-old *D. nobile* group (1.8 g/kg/day); TDN-M: middle dose of the three-year-old *D. nobile* group (3.6 g/kg/day); and TDN-H: high dose of the three-year-old *D. nobile* group (7.2 g/kg/day). ## represents *p <* 0.01 compared to the control group, * represents *p <* 0.05 compared to the diabetes model group, and ** represents *p <* 0.01 compared to the diabetes model group.

**Figure 7 molecules-29-00699-f007:**
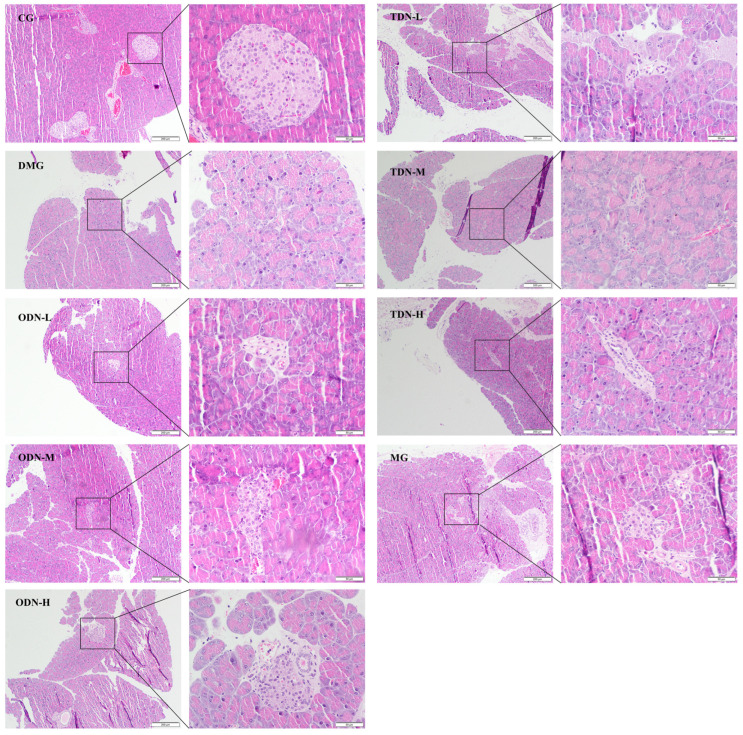
H&E staining to observe the effect of different growth years of *D. nobile* on the pancreatic tissues of diabetic mice (n = 12). The scale of the primary map is 200 μm, and the scale of the secondary enlargement is 50 μm. CG: control group; DMG: diabetes model group; ODN-L: low dose of the one-year-old *D. nobile* group (1.8 g/kg/day); ODN-M: middle dose of the one-year-old *D. nobile* group (3.6 g/kg/day); ODN-H: high dose of the one-year-old *D. nobile* group (7.2 g/kg/day); TDN-L: low dose of the three-year-old *D. nobile* group (1.8 g/kg/day); TDN-M: middle dose of the three-year-old *D. nobile* group (3.6 g/kg/day); and TDN-H: high dose of the three-year-old *D. nobile* group (7.2 g/kg/day).

**Figure 8 molecules-29-00699-f008:**
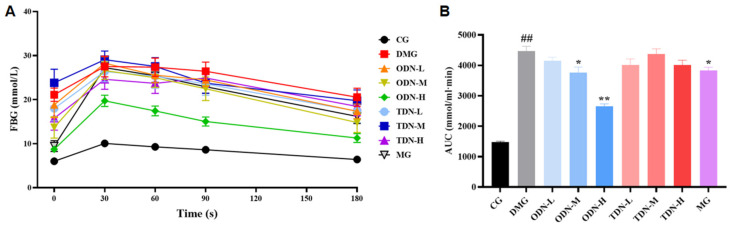
The effect of different growth years of *D. nobile* on glucose tolerance in diabetic mice (n = 12). (**A**) The blood glucose curve within 180 min after the intraperitoneal injection of glucose. The black curve represents the control group, the red curve represents the model group, the orange curve represents ODN-L, the yellow curve represents ODN-M, the green curve represents ODN-H, the cyan curve represents TDN-L, the blue curve represents TDN-M, the purple curve represents TDN-H, and the gray curve represents the positive drug group; (**B**) the statistical chart of the area under the curve. CG: control group; DMG: diabetes model group; ODN-L: low dose of the one-year-old *D. nobile* group (1.8 g/kg/day); ODN-M: middle dose of the one-year-old *D. nobile* group (3.6 g/kg/day); ODN-H: high dose of the one-year-old *D. nobile* group (7.2 g/kg/day); TDN-L: low dose of the three-year-old *D. nobile* group (1.8 g/kg/day); TDN-M: middle dose of the three-year-old *D. nobile* group (3.6 g/kg/day); TDN-H: high dose of the three-year-old *D. nobile* group (7.2 g/kg/day). ## represents *p* < 0.01 compared to the control group, * represents *p* < 0.05 compared to the model group, and ** represents *p* < 0.01 compared to the model group.

**Figure 9 molecules-29-00699-f009:**
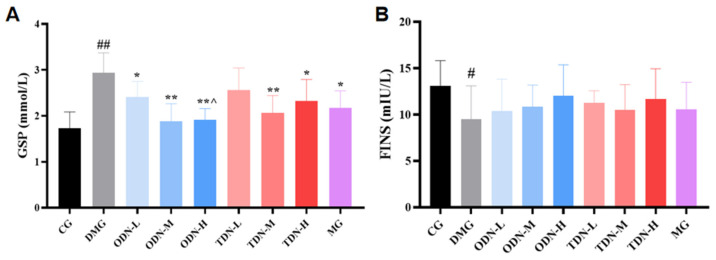
The effect of different growth years of *D. nobile* on glycated serum protein and fasting insulin levels in diabetic mice (n = 12). (**A**) Bar graph for glycated serum protein levels in each group of mice; and (**B**) bar graph for fasting serum insulin levels in each group of mice. CG: control group; DMG: diabetes model group; ODN-L: low dose of the one-year-old *D. nobile* group (1.8 g/kg/day); ODN-M: middle dose of the one-year-old *D. nobile* group (3.6 g/kg/day); ODN-H: high dose of the one-year-old *D. nobile* group (7.2 g/kg/day); TDN-L: low dose of the three-year-old *D. nobile* group (1.8 g/kg/day); TDN-M: middle dose of the three-year-old *D. nobile* group (3.6 g/kg/day); and TDN-H: high dose of the three-year-old *D. nobile* group (7.2 g/kg/day). # represents *p* < 0.05 compared to the control group, ## represents *p* < 0.01 compared to the control group, * represents *p* < 0.05 compared to the model group, and ** represents *p* < 0.01 compared to the model group.**^ represents *p <* 0.05 compared to the high-dose group of 3-year-*D. nobile*.

**Figure 10 molecules-29-00699-f010:**
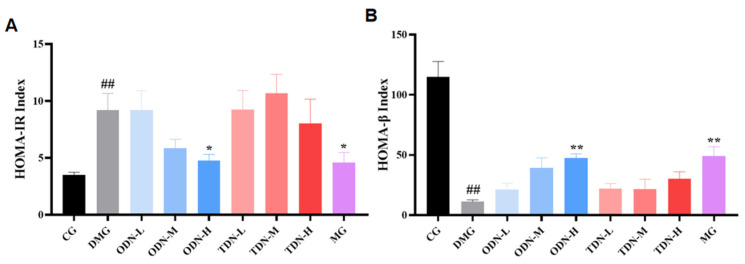
The effect of different growth years of *D. nobile* on HOMA-IR and HOMA-β in diabetic mice (n = 12). (**A**) Bar graph for steady-state insulin resistance index for each group of mice; and (**B**) bar graph for steady-state islet beta function index for each group of mice. CG: control group; DMG: diabetes model group; ODN-L: low dose of the one-year-old *D. nobile* group (1.8 g/kg/day); ODN-M: middle dose of the one-year-old *D. nobile* group (3.6 g/kg/day); ODN-H: high dose of the one-year-old *D. nobile* group (7.2 g/kg/day); TDN-L: low dose of the three-year-old *D. nobile* group (1.8 g/kg/day); TDN-M: middle dose of the three-year-old *D. nobile* group (3.6 g/kg/day); and TDN-H: high dose of the three-year-old *D. nobile* group (7.2 g/kg/day). ## represents *p* < 0.01 compared to the control group, * represents *p* < 0.05 compared to the model group, and ** represents *p* < 0.01 compared to the model group.

**Figure 11 molecules-29-00699-f011:**
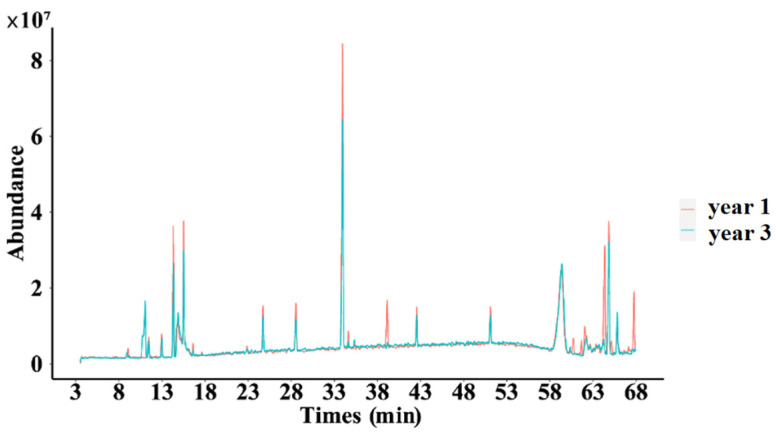
Superimposed total ion flow of secondary metabolites of *D. nobile* at different growth years. Red lines represent the 1-year *D. nobile* samples; blue lines represent the 3-year *D. nobile* samples.

**Figure 12 molecules-29-00699-f012:**
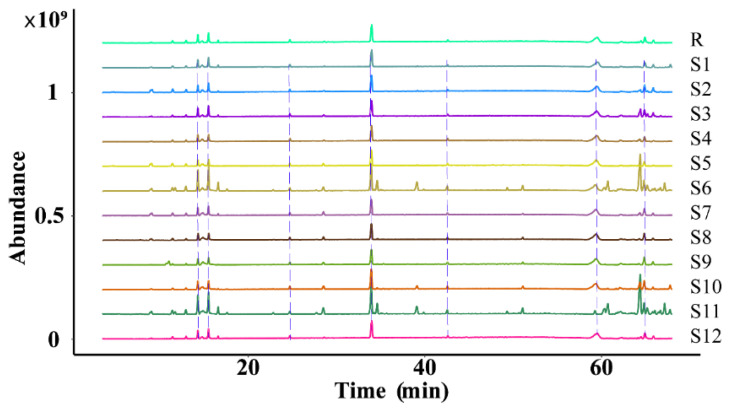
Superimposition of the fingerprint profiles of 12 batches of *D. nobile*.

**Figure 13 molecules-29-00699-f013:**
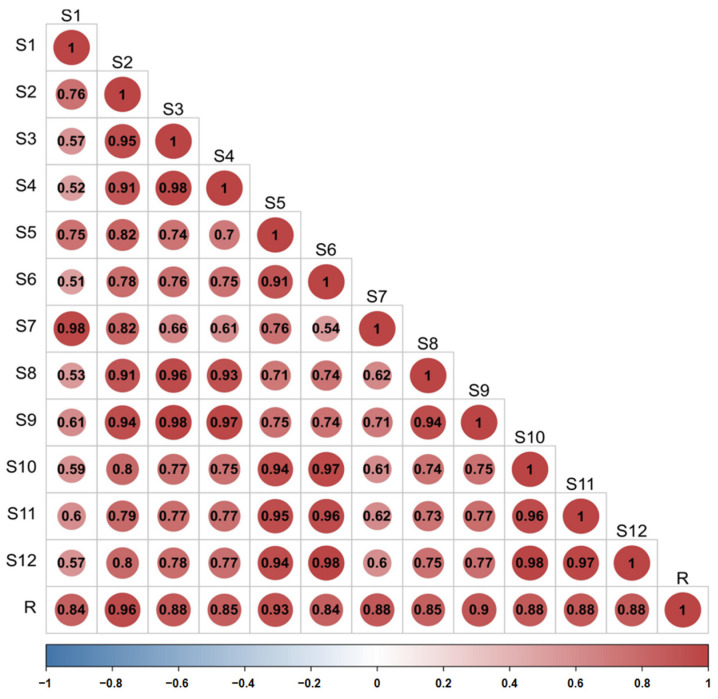
Similarity evaluation of the fingerprint profiles of 12 batches of *D. nobile* samples.

**Figure 14 molecules-29-00699-f014:**
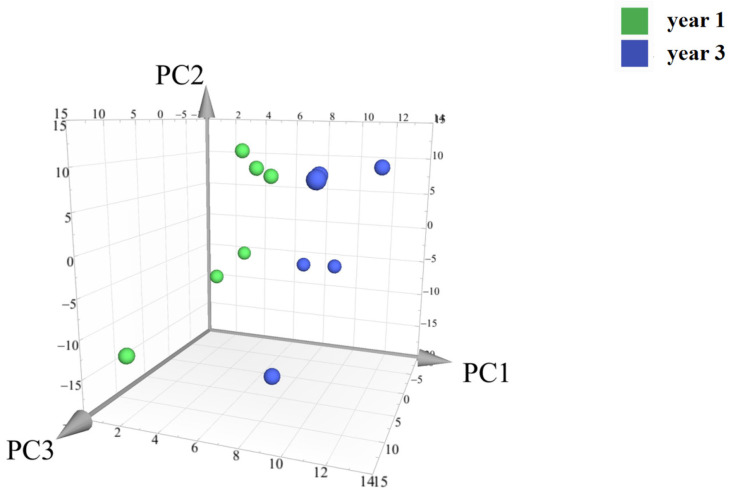
Principal component analysis of the secondary metabolites of *D. nobile* at different growth years. Notes: Each point represents a sample; the green points are samples of *D. nobile* in 1 year, and the blue ones are samples of *D. nobile* in 3 years.

**Figure 15 molecules-29-00699-f015:**
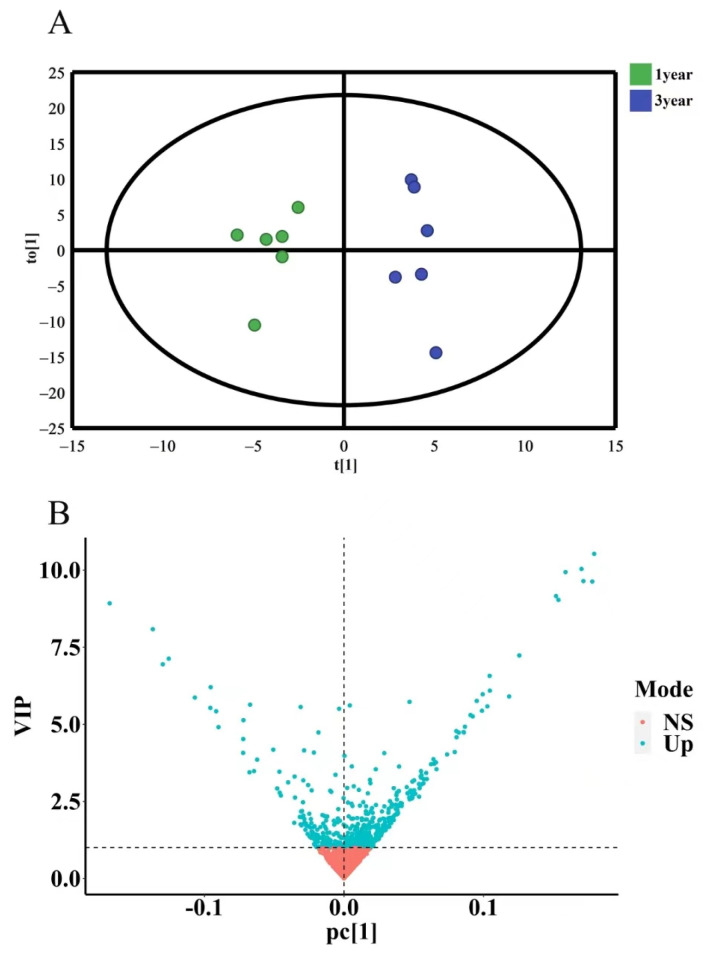
OPLS-DA analysis of secondary metabolites of *D. nobile* at different growth years. (**A**) OPLS-DA score plot, where each point represents a sample, the green points are samples of *D. nobile* in 1 year, and the blue ones are samples of *D. nobile* in 3 years; (**B**) VIP plot: the *X*-axis is PC1 values, the *Y*-axis is VIP values, and each point represents a secondary metabolite; blue points are secondary metabolites with VIP > 1, and red points are secondary metabolites with VIP < 1.

**Figure 16 molecules-29-00699-f016:**
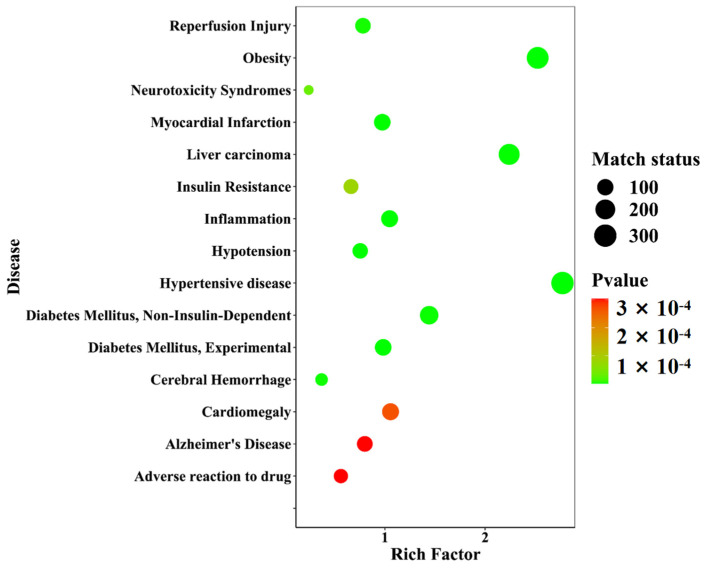
A disease enrichment bubble diagram of the differential component action targets of *D. nobile* at different growth years.

**Figure 17 molecules-29-00699-f017:**
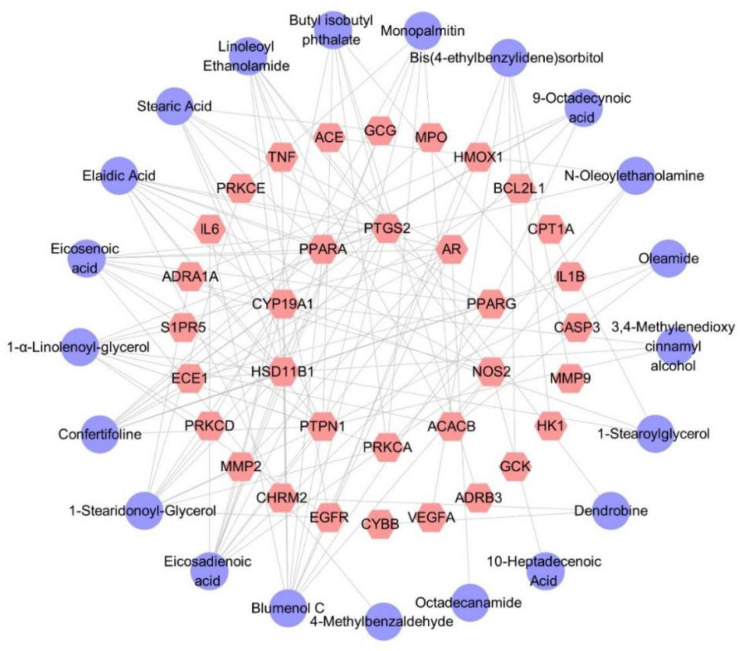
Differential component–target action network of *D. nobile* at different growth ages.The purple circles represent differential components, the pink hexagons represent the corresponding target genes.

**Table 1 molecules-29-00699-t001:** Information on the differential composition of *D. nobile* at different growth ages.

**ID**	**Name**	**Formula**	**Calc. MW**	**RT (min)**	**Class**	**VIP**
1	Dendrobine	C_16_H_25_NO_2_	263.18816	11.162	Alkaloids	2.989
2	Bis(4-ethylbenzylidene)sorbitol	C_24_H_30_O_6_	414.20360	39.268	Saccharides and alcohols	4.47091
3	1-Stearidonoyl-Glycerol	C_21_H_34_O_4_	350.24505	46.478	Lipids	1.08842
4	Confertifoline	C_15_H_22_O_2_	234.16136	46.999	Terpenoids	1.09706
5	Butyl isobutyl phthalate	C_16_H_22_O_4_	278.15114	51.276	Phenolic acids	1.9973
6	Linoleoyl Ethanolamide	C_20_H_37_NO_2_	323.28167	54.843	Alkaloids	2.42643
7	9-Octadecynoic acid	C_18_H_32_O_2_	280.24023	56.987	Lipids	2.11598
8	N-Oleoylethanolamine	C_20_H_39_NO_2_	325.29698	58.429	Alkaloids	1.40556
9	3,4-Methylenedioxy cinnamyl alcohol	C_10_H_10_O_3_	178.06252	59.276	Lignans	1.07447
10	1-α-Linolenoyl-glycerol	C_21_H_36_O_4_	352.26019	60.241	Lipids	3.91584
11	Monopalmitin	C_19_H_38_O_4_	330.27608	60.436	Lipids	2.62414
12	4-Methylbenzaldehyde	C_8_H_8_O	120.05734	60.68	Others	1.9391
13	Oleamide	C_18_H_35_NO	281.27071	61.364	Lipids	2.20184
14	Blumenol C	C_13_H_22_O_2_	210.1612	63.053	Terpenoids	1.9486
15	Elaidic Acid	C_18_H_34_O_2_	282.25506	63.506	Lipids	1.30443
16	Eicosadienoic acid	C_20_H_36_O_2_	308.27079	64.328	Lipids	1.24043
17	10-Heptadecenoic Acid	C_17_H_32_O_2_	268.24059	64.418	Lipids	1.34585
18	Octadecanamide	C_18_H_37_NO	283.28679	64.541	Alkaloids	9.10166
19	1-Stearoylglycerol	C_21_H_42_O_4_	358.30729	64.937	Lipids	1.23847
20	Stearic Acid	C_18_H_36_O_2_	284.27092	64.939	Lipids	1.11574
21	Eicosenoic acid	C_20_H_38_O_2_	310.28636	67.545	Lipids	1.1528

**Table 2 molecules-29-00699-t002:** Basic information on the *D. nobile* samples.

**Sample**	**Sample Number ***	**Picking Time**	**Latitude and Longitude**	**Growth Year**
S1	KX2021100257401	October 2021	105°47′1″ E 28°26′29″ N	One year
S2	CY2022100462301	October 2022	105°58′53″ E 28°44′24″ N	One year
S3	YJ2022100469001	October 2022	105°89′ E 28°61′ N	One year
S4	XTDPHZ2022100528001	October 2022	105°44′54″ E 28°33′37″ N	One year
S5	XTDPMX2022100528001	October 2022	105°44′54″ E 28°33′37″ N	One year
S6	ZS2022100534701	October 2022	105°76′ E 28°45′ N	One year
S7	KX2021100257403	October 2021	105°47′1″ E 28°26′29″ N	Three years
S8	CY2022100462303	October 2022	105°58′53″ E 28°44′24″ N	Three years
S9	YJ2022100469003	October 2022	105°89′ E 28°61′ N	Three years
S10	XTDPHZ2022100528003	October 2022	105°44′54″ E 28°33′37″ N	Three years
S11	XTDPMX2022100528003	October 2022	105°44′54″ E 28°33′37″ N	Three years
S12	ZS2022100534703	October 2022	105°76′ E 28°45′ N	Three years

Note: * The sampling numbering rule is the sampling base name + picking time + altitude + growth year.

**Table 3 molecules-29-00699-t003:** The gradient elution conditions for the determination of dendrobine content.

**Time (min)**	**0.1% Formic Acid in Water (A%)**	**Acetonitrile (B%)**
0	90	10
1	90	10
3	75	25
4	75	25
5	90	10
9	90	10

**Table 4 molecules-29-00699-t004:** The gradient elution conditions for screening secondary metabolites of *D. nobile*.

**Time (min)**	**0.1% Formic Acid in Water (A%)**	**Acetonitrile (B%)**
0	95	5
5	90	10
10	80	20
20	70	30
42	52	48
50	40	60
62	20	80
68	10	90
73	10	90
75	95	5
77	95	5

## Data Availability

Data are contained within the article and Appendix A.

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
