# Peer review of "Hypoglycemic Effects and Quality Marker Screening of Dendrobium nobile Lindl. at Different Growth Years"

_molecules, 2024, doi:10.3390/molecules29030699_

Round 1
Reviewer 1 Report
Comments and Suggestions for Authors
The article is about the hypoglycemic effects of Dendrobium nobile (D. nobile), a medicinal herb used in traditional Chinese medicine, and identifying potential quality markers based on their efficacy. The authors compared the ethanol extracts of one-year and three-year D. nobile on alloxan-induced diabetic mice. They found that the one-year D. nobile had a better hypoglycemic effect. They also used UPLC-QE/Orbitrap/MS to analyze the secondary metabolites of D. nobile and identified 21 differential compounds that were related to 34 key targets of diabetes. They suggested these compounds could be used as quality markers for D. nobile.
The following given questions should be answered:
1. The title could be more specific and informative, such as “Hypoglycemic effects and quality markers of Dendrobium nobile at different growth years”.
- The abstract could be more concise and avoid repeating the same information in different sections. For example, the sentence “The growth age significantly affects the composition of secondary metabolites in D. nobile.” could be omitted as it is already stated in the results and conclusions sections.
- The article provides a good background and rationale for the research question: screen for quality markers of D. nobile based on hypoglycemic efficacy. However, the literature review could be more comprehensive and include some recent and relevant references from reputable journals. That discusses the current challenges and advances in diabetes research and treatment.
4. It is advisable to highlight the significance of synthetic endeavors in the advancement of antidiabetic medication. Within this context, it is recommended to underscore the significance of iminosugars and sugar derivatives as potent antidiabetic agents. To support this assertion, referencing the subsequent pertinent articles in the introduction section is suggested: i) https://doi.org/10.1002/anie.202217809 ii) Compain, P.; Martin, O. R. Iminosugars: From synthesis to therapeutic applications; Wiley-VCH:New York, 2007; pp 187−298 and iii) https://doi.org/10.24820/ark.5550190.p011.809.
- The article describes the methods and materials used in the experiment in sufficient detail, but some information needs to be included or clarified. For example, you should specify the source and quality of the D. nobile samples, the extraction and preparation methods, the dosage and administration of the extracts, the criteria for selecting and grouping the mice, the ethical approval and animal welfare protocols, the statistical methods and software used for data analysis, and the validation and calibration of the UPLC-QE/Orbitrap/MS instrument.
- The article presents the results clearly and concisely, using tables and figures to illustrate the main findings. However, some tables and figures need to be better labeled or explained. For example, you should provide captions for each table and figure, indicate the units and scales of the axes, define the abbreviations and symbols used, and refer to them in the text. You should also avoid repeating the same information in tables and figures and choose the most appropriate format to display the data.
- The article discusses the implications and limitations of the results and compares them with previous studies. However, the discussion could be more critical and in-depth and address some of the following questions: How do the results support or contradict the existing literature? What are the possible mechanisms of action of the quality markers on the key targets related to diabetes? How generalizable and applicable are the results to other species, populations, or contexts? What are the potential benefits and risks of using D. nobile as a hypoglycemic agent? What gaps and challenges remain to be solved in future research?
- The article concludes with a summary of the main findings and contributions of the study. However, the conclusion could be more specific and concise and avoid repeating the information from the abstract or the introduction. You should also provide recommendations or suggestions for future research directions or applications based on your results.
Overall, the manuscript requires a comprehensive rewrite due to excessive length and the need for clarity in its direction.
Comments on the Quality of English LanguageAvoid repeating the same information in different sections. For example, the sentence “The growth age significantly affects the composition of secondary metabolites in D. nobile.” could be omitted as it is already stated in the results and conclusions sections.
Author Response
Dear Reviewer,
We have already revised the manuscript (Manuscript ID: molecules-2820845) according to the reviewers’ comments.
We have uploaded the revised manuscript with track and later we will upload the other one is clean version.
We have gone through the all text, almost re-written most of them. We also answered all questions from reviewers at the end of this letter.
We hope the revised manuscript meet the requirements of this journal. Please feel free to contact us if the manuscript still needs any further revision.
Answers are as follows:
1.The title could be more specific and informative, such as“Hypoglycemic effects and quality markers of Dendrobium nobile at different growth years”.
Answer:Yes, your suggestion is more specific and informative, we have already changed the title into “Hypoglycemic effects and quality markers screening of Dendrobium nobile at different growth years”, thanks.
2. The abstract could be more concise and avoid repeating the same information in different sections. For example, the sentence “The growth age significantly affects the composition of secondary metabolites in D. nobile.” could be omitted as it is already stated in the results and conclusions sections.
Answer: We have already gone through the abstract, and omitted redundant information. Thanks
3.The article provides a good background and rationale for the research question: screen for quality markers of D. nobile based on hypoglycemic efficacy. However, the literature review could be more comprehensive and include some recent and relevant references from reputable journals. That discusses the current challenges and advances in diabetes research and treatment.
Answer: We have already revised the first paragraph of the introduction section. Thanks for your kindly suggestion.
4.It is advisable to highlight the significance of synthetic endeavors in the advancement of antidiabetic medication. Within this context, it is recommended to underscore the significance of iminosugars and sugar derivatives as potent antidiabetic agents. To support this assertion, referencing the subsequent pertinent articles in the introduction section is suggested: i) https://doi.org/10.1002/anie.202217809IF: 16.6 Q1 ii) Compain, P.; Martin, O. R. Iminosugars: From synthesis to therapeutic applications; Wiley-VCH:New York, 2007; pp 187−298 and iii) https://doi.org/10.24820/ark.5550190.p011.809IF: 0.9 Q4 .
Answer: We have already included the information from your recommended references and cited them in our revised manuscript.
5.The article describes the methods and materials used in the experiment in sufficient detail, but some information needs to be included or clarified. For example, you should specify the source and quality of the D. nobile samples, the extraction and preparation methods, the dosage and administration of the extracts, the criteria for selecting and grouping the mice, the ethical approval and animal welfare protocols, the statistical methods and software used for data analysis, and the validation and calibration of the UPLC-QE/Orbitrap/MS instrument.
Answer:
Regarding the source and quality of D. nobile, the samples were collected from various planting areas in Chishui region of Guizhou province in China. The quality was identified by Professor Faming Wu from School of Pharmacy of Zunyi Medical University according to China Pharmacopeia. We have already included this information in the section “Materials and Methods / 4.1 Plant Materials and pre-treatment”. The information in detail could be found in the Table 4.
The extraction and preparation process of D. nobile was as follows: After pre-processing, D. nobile samples were accurately weighed and added to a 95% ethanol-water solution. The mixture was then heated and refluxed for three times, filtered while hot and concentrated at low pressure, subsequently subjected to vacuum freeze-drying before storing. The aforementioned process was also included in the section “Materials and Methods / 4.1 Plant Materials and pre-treatment”.
The dosage of D. nobile referred to the criteria described in China Pharmacopeia. The low dosage was equal to that indicated in China Pharmacopeia. The medium dosage was two folds of the low dosage, while the high dosage was 4 folds of the low dosage. This information has already been included in the second paragraph of the section “4.4.1. Modeling and Administration”
A total of 132 male C57BL/6J mice, aged 7-8 weeks and weighing 20-22 g were used in our study. After modeling, mice receiving alloxan with FBG levels higher than 11.1 mmol/L were randomly divided into the model control and treatment groups, while the mice receiving saline were randomly selected in the normal control group. These information was included in the section “4.4.1. Modeling and Administration”
This experiment has been approved by the Animal Experiment Ethics Committee of Zunyi Medical University(lience number:[2021] 2-606), and were described in the section “4.4.1. Modeling and Administration”
The statistical methods were described in section “4.7. Statistical Methods”
The software used in the present study were mentioned in the correspondent section, and collectively described in the section “Main Software and Database” of the supplemental files.
The UPLC-QE/Orbitrap/MS instrument was calibrated and validated by using dendrobine as the reference compound. The methodology data was shown in the first section “Methodological validation” of the supplemental files.
6.The article presents the results clearly and concisely, using tables and figures to illustrate the main findings. However, some tables and figures need to be better labeled or explained. For example, you should provide captions for each table and figure, indicate the units and scales of the axes, define the abbreviations and symbols used, and refer to them in the text. You should also avoid repeating the same information in tables and figures and choose the most appropriate format to display the data.
Answer: We have already illustrate all abbreviations and symbols in the figure legends and correspondent text. We also visualized the data in the tables to avoid redundant information. Thanks.
7.The article discusses the implications and limitations of the results and compares them with previous studies. However, the discussion could be more critical and in-depth and address some of the following questions: How do the results support or contradict the existing literature? What are the possible mechanisms of action of the quality markers on the key targets related to diabetes? How generalizable and applicable are the results to other species, populations, or contexts? What are the potential benefits and risks of using D. nobile as a hypoglycemic agent? What gaps and challenges remain to be solved in future research?
Answer: We have polished the discussion according to your guidance. You can find it in the beginning of the discussion section. Thanks a lot.
8.The article concludes with a summary of the main findings and contributions of the study. However, the conclusion could be more specific and concise and avoid repeating the information from the abstract or the introduction. You should also provide recommendations or suggestions for future research directions or applications based on your results.Overall, the manuscript requires a comprehensive rewrite due to excessive length and the need for clarity in its direction.
Answer: We have already re-written the conclusion section and hope it meets your requirements. Thanks for your guidance.

Reviewer 2 Report
Comments and Suggestions for Authors
The sentence in line Line 86 is not required.
"Determination of Dendrobin Content by LC-MS" - reference for the methodology is missing. If the method was developed and validated for this study, specify the validation parameters.
The same comment for the part - Determination of the Content of Sesquiterpene Glycosides by LC-MS - methodology reference or validation protocol and results.
Lines 127, and 128 are missing a protocol reference. If it was not done according to some protocol, and based on which the applied dose was calculated.
How is the glucose dose calculated for the Intraperitoneal Glucose Tolerance Test?
Technical correction
Line 284, 303, and 456 D. nobile write in italic.
Review the references once again - Lines 593, 598, 600, 625, etc. contains month and date
Author Response
Dear Reviewer,
We have already revised the manuscript (Manuscript ID: molecules-2820845) according to the reviewers’ comments.
We have uploaded the revised manuscript with track and later we will upload the other one is clean version.
We have gone through the all text, almost re-written most of them. We also answered all questions from reviewers at the end of this letter.
We hope the revised manuscript meet the requirements of this journal. Please feel free to contact us if the manuscript still needs any further revision.
Answers as follows:
1. The sentence in line Line 86 is not required.
Answer: We have already omitted this sentence. Thanks.
2. "Determination of Dendrobin Content by LC-MS" - reference for the methodology is missing. If the method was developed and validated for this study, specify the validation parameters.
Answer: The UPLC-QE/Orbitrap/MS instrument was calibrated and validated by using dendrobine as the reference compound. The methodology data was shown in the section “Methodological validation” and “Methodological results” of the supplemental files.
3.The same comment for the part - Determination of the Content of Sesquiterpene Glycosides by LC-MS - methodology reference or validation protocol and results.
Answer: The UPLC-QE/Orbitrap/MS instrument was calibrated and validated by using Longdanxieganin as the reference compound.And the data was shown in the section “2.1 Chromatogram of the determination the content of dendrobine and sesquiterpene glycosides” of the supplemental files.
4.Lines 127, and 128 are missing a protocol reference. If it was not done according to some protocol, and based on which the applied dose was calculated.
Answer: The dosage of D. nobile referred to the criteria described in China Pharmacopeia. The low dosage was equal to that indicated in China Pharmacopeia. The medium dosage was two folds of the low dosage, while the high dosage was 4 folds of the low dosage. This information has already been included in the second paragraph of the section “4.4.1. Modeling and Administration”. And the dosage of metformin is refered by the subsequent report.
Ansari A, Bose S, Lim SK, Wang JH, Choi YH, Kim H. Combination of Scutellaria baicalensis and Metformin Ameliorates Diet-Induced Metabolic Dysregulation in Mice via the Gut-Liver-Brain Axis. Am J Chin Med. 2020;48(6):1409-1433. doi: 10.1142/S0192415X2050069.
5.How is the glucose dose calculated for the Intraperitoneal Glucose Tolerance Test?
Answer: The dosage of glucose is refered by the subsequent report.
Andrikopoulos S, Blair AR, Deluca N, Fam BC, Proietto J. Evaluating the glucose tolerance test in mice. Am J Physiol Endocrinol Metab. 2008 Dec;295(6):E1323-32. doi: 10.1152/ajpendo.90617.2008. Epub 2008 Sep 23. PMID: 18812462.
6. Line 284, 303, and 456 D. nobile write in italic.
Answer: We have already revised these mistakes. Thanks for your kindly suggestion.
7.Review the references once again - Lines 593, 598, 600, 625, etc. contains month and date
Answer: We have already gone through the references, and corrected mistakes.
